

# Myocardial contrast echocardiography assessment of mouse myocardial infarction: comparison of kinetic parameters with conventional methods

Nicholas T. Thielen[1], Adison A. Kleinsasser[2] and Jessica L. Freeling[3]

[1] Sanford School of Medicine, University of South Dakota, Vermillion, South Dakota, United States
[2] Research Computing Group, University of South Dakota, Vermillion, South Dakota, United States
[3] Basic Biomedical Sciences, University of South Dakota, Vermillion, South Dakota, United States

Corresponding author
Jessica L. Freeling, jessica.freeling@usd.edu

## ABSTRACT

This study explores the use of a minimally invasive assessment of myocardial infarction (MI) in mice using myocardial contrast echocardiography (MCE). The technique uses existing equipment and software readily available to the average researcher. C57/BL6 mice were randomized to either MI or sham surgery and evaluated using MCE at 1- or 2-weeks post-surgery. Size-isolated microbubbles were injected via retro-orbital catheter where their non-linear characteristics were utilized to produce the two-dimensional parameters of Wash-in-Rate and the Peak Enhancement, indicative of relative myocardial perfusion and blood volume, respectively. Three-dimensional cardiac reconstructions allowed the calculation of the Percent Agent, interpreted as the vascularity of the entire myocardium. These MCE parameters were compared to conventional assessments including M-Mode, strain analysis, and 2,3,5-Triphenyltetrazolium chloride (TTC) staining. Except for the Wash-in-Rate 2-week cohort, all MCE parameters were able to differentiate sham-operated versus MI animals and correlated with TTC staining ($P < 0.05$). MCE parameters were also able to identify MI group animals which failed to develop infarctions as determined by TTC staining. This study provides basic validation of these MCE parameters to detect MI in mice complementary to conventional methods while providing additional hemodynamic information in vivo.

## INTRODUCTION

Cardiovascular diseases are the leading cause of death globally, affecting an estimated 17.9 million people in 2016 (*WHO, 2017*). With such a high burden on society, there is an increasing demand for basic translational cardiovascular research to identify mechanisms and treatment options. Due to the high incidence of coronary artery disease, advances in pre-clinical research methods are important, particularly in relevant and widely applied

animal models. The murine model is widely used (*Michael et al., 1995*) for human cardiovascular disease because of the physiological similarities, and rapidly expanding availability, of genetic modification tools (*Scherrer-Crosbie & Thibault, 2008*; *Wang et al., 2006*). This presents the need for physiologically relevant methods to accurately assess myocardial infarction (MI) in mice.

Currently validated methods of assessing MI with functional echocardiographic data offer a variety of limitations which could be improved upon with the addition of a perfusion imaging modality (*Bhan et al., 2014*; *Chen et al., 2016*). Recent developments into the understanding of coronary microvascular disease further emphasize the need for accurate perfusion imaging as functional imaging may not detect coronary microvascular disease (*Taqueti & Di Carli, 2018*). Furthermore, coronary microvascular disease may play a role in stunned and hibernating myocardium and require the detection of preserved myocardial perfusion to make this diagnosis (*Kloner, 2020*). The most commonly utilized strain for MI research, the C57BL/6 mouse, is known to exhibit multiple left anterior descending artery (LAD) phenotypic patterns, making consistency of MI pattern, and thus infarct area, particularly challenging (*Ahn et al., 2004*). Therefore, the development of a technique that can accurately assess parameters of vascularity of the myocardial wall would provide researchers with a more robust quantification of MI. This would enable the testing of longitudinal treatment paradigms and provide further insight into the effects of both epicardial and micro-vascular disease on health outcomes.

Considering the availability and minimally invasive nature of ultrasound, myocardial contrast echocardiography (MCE) may be a useful modality to assess myocardial perfusion. It is currently a mobile and widely available imaging modality that can assess myocardial blood flow and myocardial blood volume in vivo without ionizing radiation (*Kaul, 2008*; *Verkaik et al., 2018*). MCE has been established in humans and is typically performed alongside 2D B-Mode imaging (*Porter et al., 2020*). Techniques in patients often utilize bolus injection or continuous infusions of microbubbles that require a high mechanical index burst sequence and subsequent replenishment analysis (*Averkiou et al., 2020*). Indeed, the use of MCE in dogs and rats to measure visual perfusion defects have been validated (*Chen et al., 2009*; *Dourado et al., 2003*). These techniques benefit from the relatively large size of the animal allowing enhanced visualization of microbubbles in the myocardium but entail a high degree of subjectivity in assessing the perfusion defect.

The use of MCE to evaluate mouse MI is well established. An early study validated MCE in mice with a correlation of myocardial video intensity to Evans blue staining (*Scherrer-Crosbie et al., 1999*). More recent advances involve creating a 3D rendering with MCE which could visually be compared with postmortem histology staining (*French et al., 2006*). Another method of quantifying MI with MCE has been performed by plotting the video intensity of a one-pixel-wide line against time to indicate the perfusion of the myocardium (*Alvarez et al., 2018*). These methods continue to validate the use of MCE in mice; however, they offer multiple areas that can be improved upon. Notably, none of them use an objective parameter to quantify the non-linear contrast signal given off from the myocardium.

The use of MCE in mice presents several issues, most of which involve the inherently small size of the heart. In mice, high-frequency ultrasound is required for imaging to facilitate sufficient resolution for accurate quantification, without excessive penetration. The higher frequencies required for mice therefore reduces both the ability to utilize harmonic imaging with microbubbles (*Scherrer-Crosbie & Thibault, 2008*), and the ability to penetrate through the opacified left ventricular (LV) chamber to visualize the posterior wall. These were reported to be resolved with methods that involved lodging large size isolated microbubbles within the myocardium after the lumen cleared (*Kaufmann et al., 2007*); however, this technique has not yet been reproduced. Most published techniques rely on subjective measurements of video/pixel intensity rather than measuring non-linear contrast (NLC) signal. Also, the protocols involved in these techniques and post-imaging analyses rely on complex self-made mathematical models or software programs that are not readily available, making them too time consuming and difficult to be performed in an efficient manner for the average cardiovascular researcher.

Here, we evaluate a comprehensive method that can identify and further characterize mouse MI using 2D and 3D NLC MCE. We utilized a commonly available ultrasound machine and post-imaging software, employed the most widely used anesthesia regimen, and optimized the workflow, all of which will enable researchers to perform these techniques proficiently. Moreover, this technique detects in vivo MI in 3D by reporting the Percent Agent (PA), taking advantage of the non-linear nature of the microbubbles to report an objective value reflecting hemodynamic real-time information of the entire myocardium. Importantly, these MCE parameters were validated and correlated both with 2,3,5-Triphenyltetrazolium chloride (TTC) staining of ex vivo myocardium and in vivo conventional functional ultrasound measurements. Thus, this minimally invasive technique uses size-isolated microbubbles (SIMB) and NLC MCE to identify MI and provide relevant hemodynamic information about myocardial perfusion, enabling enhanced characterization of the mouse MI model.

## MATERIALS AND METHODS

### Animal model

The University of South Dakota Institutional Animal Care and Use Committee approval was obtained for all experiments, IACUC # 12-12-17-20D. We used a total of 28 female C57BL/6NHsd mice in this study (Envigo, Denver, CO, USA) as this is the most widely utilized strain in cardiovascular research. All animals were housed in an American Association for Accreditation of Laboratory Animal Care accredited facility kept within appropriate temperature and humidity levels and exposed to 12/12 light/dark cycle. Mice were housed in microisolator cage racks, fed standard rodent chow ad libidum, and provided with environmental enrichment in the form of cotton bedding squares and plastic huts. Under general isoflurane anesthesia, 12 animals were randomly selected for sham surgery and 16 randomly selected for permanent LAD coronary artery ligation using an established surgical technique (*Tarnavski et al., 2004*). Buprenorphine SR LAB (Zoopharm, Laramie, WY, USA) 1.0 mg/kg SQ given once, lasting 72 h was given as analgesia prior to the animal regaining consciousness. Animals were monitored twice daily

for signs of distress, pain, or dehydration in the three days after surgery. While the IACUC protocol included provision to euthanize animals if they exhibited any of these symptoms, no mice in this study met criteria for euthanasia. However, two mice died during surgery due to hemmorhage and six mice died in the immediate post-surgical period of acute infarction. These losses were considered to be within normal failure rates for this procedure. Mice were evaluated using imaging as separate one-week and two-week post-infarction surgery cohorts so that comparisons could be made with conventional ex vivo staining methods. Upon completion of the imaging session, animals were never allowed to regain consciousness and further induced with 4–5% isoflurane anesthesia to ensure maximal anesthetic plane. Tissues were then collected by cardiac excision for further analysis.

The mice were prepared for imaging by induction and maintenance with 4% isoflurane and 1.25–1.75% isoflurane in 100% oxygen, respectively. For anesthesia maintenance, the head was placed in an anesthesia mask with a modified notched nosecone to allow access to the eye for retro-orbital injection. A commercial depilatory cream was used to remove fur from the anterior chest. Mice were placed in the supine position on a mobile heated imaging platform maintained at 37 °C, with the extremities taped to electrocardiogram (ECG) electrodes for heart rate and respiratory monitoring. Extreme care was taken to ensure that each mouse was at minimal anesthetic depth to enable maximal physiological function. The ears were also taped to the platform to decrease movement during the injection of microbubbles via the retro-orbital route.

A previously reported retro-orbital intravenous injection technique was used due to its proven efficacy and ease of access (*Wang et al., 2015a*; *Yardeni et al., 2011*), but with further refinement. A catheter was fashioned from a Terumo 0.5 inch 27G butterfly catheter by removing the existing tubing and replacing it with 10 cm of polyethylene 20 tubing. After catheter assembly, and before introduction to the animal, flushing with 0.9% saline was performed to ensure the smooth delivery of microbubbles without resistance to prevent unintended rupture upon delivery to the mouse. The modified nosecone enabled visualization and access to the eye during catheter placement and ear taping enabled stability to maintain catheter patency. Gentle pressure was applied at the base of the skull to create mild protrusion of the eye. The needle was inserted with the bevel towards the globe, into the medial canthus at approximately the lacrimal duct. The catheter was then stabilized with tape and flushed with 50 μl saline to test patency and venous access. Once the retro-orbital catheter was placed, extreme care was taken to prevent movement of the animal during subsequent imaging.

## Microbubbles

The bolus perfusion model was chosen for this study due to the simplicity of the technique and smaller total volumes of fluid delivered to the animal relative to the destruction-replenishment model. After trials of various sizes of neutral size-isolated microbubbles (nSIMB) 5–8, 4–5, and 3–4 μm, it was determined that the 3–4 μm SIMB (Advanced Microbubble Laboratories LLC, Boulder, CO, USA) provided an optimal enhancement of

echogenicity. They were diluted to $4 \times 10^8$ bubbles/ml, for a total of $2 \times 10^7$ bubbles injected in a 50 µl bolus, followed by a 50 µl bolus saline flush.

For comparison, we also tested size 3–4 µm cationic SIMB (cSIMB) due to their reported ability to adhere to the negatively charged vascular endothelium (*Sirsi et al., 2012*). This was done to attempt enhancement of the myocardium while the ventricular lumen was cleared of microbubbles. These were administered in the same dilution as the nSIMB.

## Imaging overview

We used the Vevo 2100 High-Resolution Micro-Ultrasound System and associated Vevo LAB software v3.2 (FUJIFILM VisualSonics, Toronto, ON, Canada). All data was stored in the central university maintained server, backed up twice daily. For comparison of our proposed MCE technique with the commonly performed functional imaging methods used for mice, we collected B-Mode and M-Mode parasternal long axis (PLAX) and parasternal short axis (PSAX) images from each animal. For further comparison to MCE, the collected B-Mode images were also evaluated using speckle tracking echocardiography (STE) for the functional evaluation of cardiac strain. This software uses a Lagrangian model of strain (*Theodoropoulos & Xu, 2008*).

For MCE, we determined that the 21 MHz MS250 linear array probe provided the best combination of resolution and penetration. Additionally, this probe has the frequency necessary to perform the NLC function and detection of the SIMB. A 3D motor controlled the probe and created 3D images with a step size of 0.07 mm and a scan length of 10.06 mm, recording ~286 frames. These were gated to both respirations and the ECG. End-systole was chosen to acquire images due to the more consistent signal intensity and the increased ability to visualize and trace the myocardium (*Verkaik et al., 2018*). All images were stored for later evaluation. Image analysis was performed by individuals blinded to the treatment (MI versus sham-operated) and type of bubbles (nSIMB versus cSIMB).

## Sequence of image acquisition

Once the retro-orbital catheter was placed and the animal was prepared for imaging, we first acquired standard 2D B-Mode and M-Mode PLAX images with the cursor placed at approximately the mid-papillary level. The transducer was then turned to acquire PSAX images at the mid-papillary level in 2D B-Mode and M-Mode. Images were acquired with settings of frequency 21 MHz, power 100%, and a dynamic range of 60 dB. To ensure adequate inclusion of the entire heart we used a depth of 14 mm and a width of 14.04 mm, which resulted in a frame rate of 167 frames/sec. M-Mode images were acquired with similar settings, except frame rate was 20 frames/s, gate depth 12.92 mm, and size 7 mm. Time gain compensation controls were set to optimize image quality and recorded as a preset to be used with all imaging.

After standard 2D B-Mode and M-Mode PLAX/PSAX images were acquired, the MCE data collection sequence was initiated. Before the delivery of microbubbles, a pre-scan was obtained. To do this, NLC Mode was initiated along with the simultaneous

activation of 3D Mode (3D+NLC). During 3D+NLC Mode, gating was applied and a 3D pre-scan (i.e., pre-injection) of ~286 images was acquired in the transverse PSAX plane along the entire length of the heart for later background subtraction. The settings in 3D Mode were 18 MHz, power 10%, and dynamic range 40 dB. Contrast and 2D gain were set at 24 dB and 21 dB, respectively.

After completion of the 3D+NLC pre-scan, the transducer was then rotated back to the sagittal plane to obtain a PLAX view and the machine was set to 2D NLC Mode to acquire the MCE arrival kinetic data from the initial microbubble injection. The timing of image acquisition was coordinated with the injection of the microbubbles so that an adequate baseline image and plateau of the kinetics of the microbubbles arriving could be recorded, ~7-s pre-injections, and ~20-s total. Prior to locking in the image acquisition regimen, we had tested microbubble stability by collecting additional post-injection PLAX 2D NLC images at ~1–2 min, ~8-min, and ~12-min post microbubble injection. From this, we determined that the stability of circulating microbubbles had reached a plateau at ~1–2 min post-microbubble injection, were still present at 8-min post-injection, but were largely cleared by 12-min. Therefore, to facilitate comparison of nSIMB and cSIMB, another 2D NLC PLAX image was obtained at 8-min post-injection. Additionally, to adequately capture the 3D+NLC post-scan while the microbubbles were stable and still adequately in circulation, at ~1–2 min, we returned the probe to the transverse PSAX and recorded the 3D+NLC post-scan. The collection of a 3D+NLC pre-scan and post-scan facilitated background subtraction to be performed upon image analysis, allowing quantification of the total microbubble percent agent (PA) of the myocardium while the microbubbles were stably circulating.

After this entire protocol was completed with nSIMB, two high mechanical index burst sequences were used to ensure full clearance of the animal of microbubbles. The entire protocol above was then repeated in the same animal with the size 3–4 μm cSIMB. As explained, the 8-min PLAX 2D NLC scan was utilized to compare the neutral versus cationic microbubble myocardial lodging. None of the protocols described have been registered.

## IMAGE ANALYSIS

### Conventional echocardiogram assessments

M-Mode and 2D B-Mode images were loaded onto Vevo LAB 3.2 software. Anterior and posterior LV walls of PLAX and PSAX M-Mode images were defined using the trace technique and software algorithms calculated an ejection fraction (EF). VevoStrain software, which enables STE analysis from 2D B-Mode images, was used to calculate parameters including an additional EF, global longitudinal strain (GLS), and global circumferential strain (GCS) using the 2D B-Mode PLAX and PSAX images.

### 2D NLC assessments

The kinetics of microbubble arrival as well as the stability of microbubbles at 8-min post-injection was evaluated. 2D NLC microbubble arrival kinetics enabled evaluation of parameters of relative blood perfusion and volume. The 8-min post-injection 2D NLC data

was evaluated to assess the stability of the nSIMB versus the cSIMB to determine if lodging of microbubbles in the myocardium could be detected. To quantify these parameters, the 2D NLC video images, which were collected with constant gain and dynamic range, were loaded into VevoCQ software and images were corrected for motion artifact. A region of interest (ROI) was traced along with the endo- and epicardium in PLAX so the whole 2D myocardium was included. The ROI was traced while microbubbles were present in the myocardium, taking advantage of the enhanced delineation of the endocardium. From these 2D NLC videos, the software calculated a wash-in-rate (WiR) representing relative blood perfusion, and peak enhancement (PE) representing relative blood volume present in the myocardium (*Greis, 2011*). The software calculated these from the maximum slope and plateau of the microbubble arrival kinetics to the myocardium, respectively (*VisualSonics, 2010*) (Fig. 1). For cSIMB and nSIMB comparisons, the ROI was copied and pasted onto the 8-min image to ensure that same area of the myocardium was under consideration and to further understand the characteristics of the differentially charged microbubbles through this time frame.

## 3D PA assessments

3D PA data is obtained from a series of ~300 individual 2D image frames that are compiled to form a 3D image. 3D scan collection time typically took less than three minutes. Microbubbles were likely stably circulating during the scan as we showed that microbubbles were present for at least eight minutes (Table 1). To facilitate analysis of the 3D+NLC data to calculate the PA, the pre-scan and ~1–2-min post-scan 3D+NLC images were loaded into the Vevo LAB 3D software. The ROIs were traced in PSAX on the ~1–2-min post-scan image over the epicardium and endocardium, so the entire 3D myocardium was included, using the multi-slice tracing technique (Fig. 2). ROIs were traced individually, every 5-10 frames, from the base to the apex of the heart to provide a smooth transition from each tracing. The 3D+NLC pre-scan (pre-injection) images were used as background subtraction from the post-scan (post-injection) images to calculate a percent agent (PA). This was calculated from the number of pixels within a defined volume, which are associated with the non-linear contrast signal given off by microbubbles. The PA indicates the relative vascularity of the entire defined myocardium in 3D.

## TTC evaluation

For TTC staining, the ex vivo myocardium was sliced into four sections in the transverse plane for comparison to the MCE parameters. TTC was used due to its long history as the gold standard of quantifying MI ex vivo (*Fishbein et al., 1981*; *Klein et al., 1981*). It was also used as the sole staining technique for simplicity purposes and because it indirectly indicates perfusion in a permanent coronary ligation model. Within an hour of completion of ultrasound imaging, the mice were deeply sedated using 4–5% isoflurane in 100% oxygen and the heart was excised. The heart was placed in a solution containing 56 mg of heparin/L in phosphate buffered saline and allowed to pump thoroughly. The aorta of the

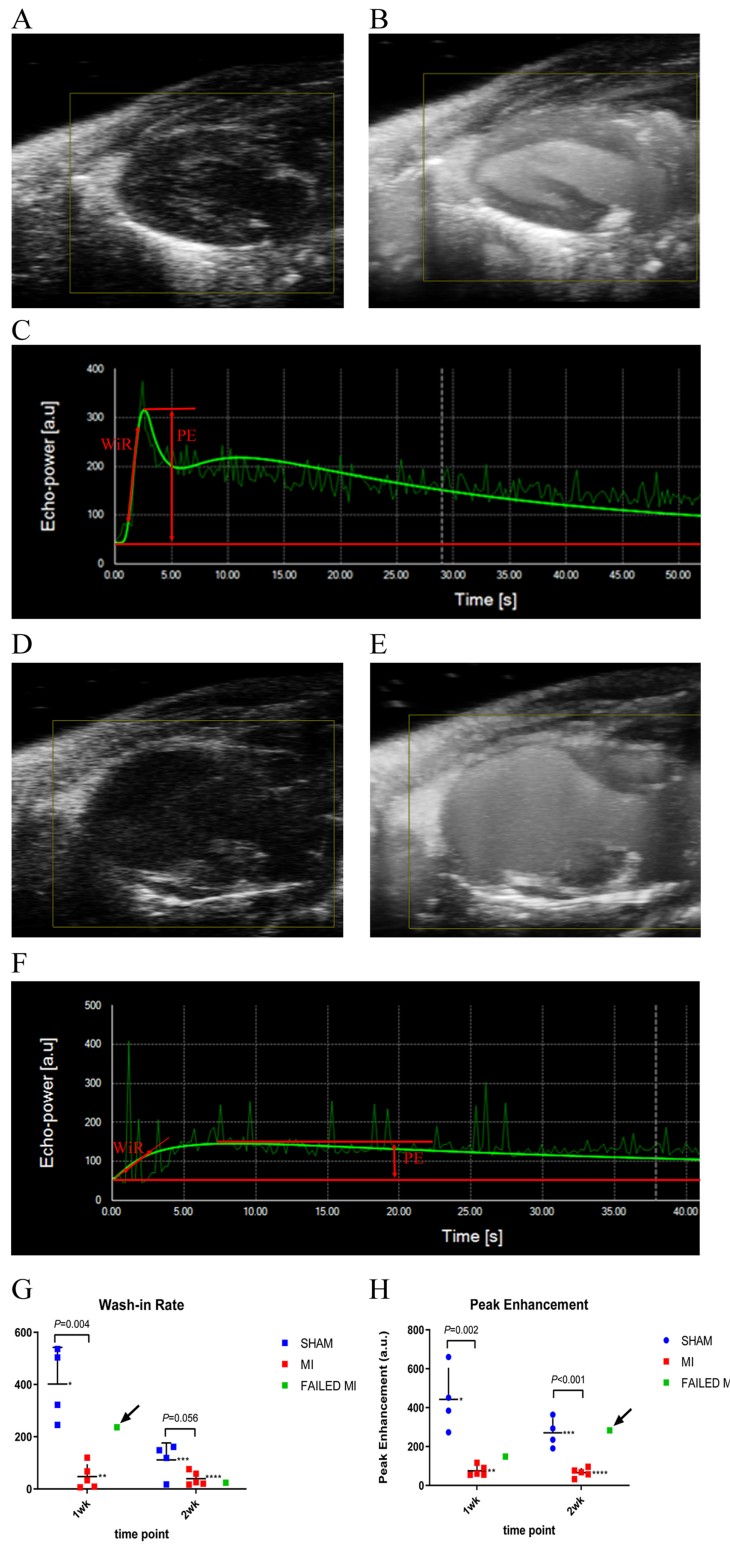

**Figure 1 Example ultrasound images of the 2D PLAX view of the left ventricle at multiple stages during the process of microbubble injection.** (A) 2D B-Mode PLAX of the left ventricle pre-microbubble injection in a sham-operated animal. (B) 2D PLAX post-microbubble injection of sham-operated animal as an MIP image, which shows the total accumulation of microbubbles over a

**Figure 1 (continued)**
period of time to better display the distribution of microbubbles in the myocardium. (C) Kinetic graph of the non-linear contrast (NLC) signal of microbubble arrival and plateau in VevoCQ software of a sham-operated animal upon microbubble injection. Locations where WiR and PE were derived from are approximated with red markings. (D) 2D B-Mode PLAX of the left ventricle pre-microbubble injection in an MI animal. (E) 2D B-Mode PLAX post-microbubble injection of an MI animal as MIP image. (F) Kinetic graph of the NLC signal of microbubble arrival and plateau in VevoCQ software of MI animal upon microbubble injection. Locations where WiR and PE were derived from are approximated with red markings. (G) Scatter plot of individual data points for the WiR calculated from the slope of the microbubble arrival kinetics in panels C and F. Data represents 1-week and 2-week cohorts of sham-operated animals (blue, $n = 4$), MI animals (red, $n = 5$), and failed MIs (green) with arrows indicating if they were able to be differentiated from the MI cohort. The failed MIs were excluded from mean data calculations. [*]401.86 ± 140.44, [**]70.32 ± 95.92, [***]111.30 ± 65.12, [****]39.27 ± 26.07. (H) Scatter plot of individual data points for the PE calculated from the plateau of the microbubble arrival kinetics in panels C and F. Data represents sham-operated animals (blue, $n = 4$), MI animals (red, $n = 5$), and failed MIs (green) with arrows indicating if they were able to be differentiated from the MI cohort. The failed MIs were excluded from mean data calculations. [*]442.29 ± 163.00, [**]75.08 ± 27.23, [***]270.39 ± 75.01, [****]65.68 ± 23.61. MCE, myocardial contrast echocardiography; MI, myocardial infarction; PLAX, parasternal long axis; MIP, maximum intensity projection; NLC, non-linear contrast; PE, peak enhancement; WiR, wash-in-rate.

**Table 1 Comparison of potential myocardial lodging of neutral size-isolated microbubbles (nSIMB) and cationic size-isolated microbubbles (cSIMB) using the MCE parameters PE and WiR.**

|  | nSIMB | cSIMB | *P* |
|---|---|---|---|
| PE Arrival [a.u.] | 356.34 ± 149.14 | 354.58 ± 185.17 | 0.98 |
| PE 8-min [a.u.] | 49.35 ± 47.52 | 24.64 ± 19.28 | 0.19 |
| WiR [a.u.] | 256.58 ± 185.45 | 378.08 ± 334.31 | 0.38 |

**Notes:**
Only sham-operated animals were included. 1- and 2-week cohorts were combined for $n = 8$ for each cohort of microbubble type.
PE Arrival, peak enhancement plateau shortly after microbubble arrival to the myocardium; PE 8-min, peak enhancement plateau 8-min after microbubble arrival; WiR, wash-in-rate; MCE, myocardial contrast echocardiography; a.u., arbitrary unit.

heart was cannulated with a 24G mouse gavage needle cannula and tied in place with 6-0 silk suture (*Bohl et al., 2009*). The heart was then gently flushed with 2–3 ml of the phosphate buffered saline/heparin, taking care to ensure that the vasculature of the heart was effectively cannulated and not the ventricular lumen. Using a syringe pump, the heart was flushed at 67 µl/min for 15 min with 1% TTC solution (0.1 g/10 ml per vial) (#17779, Sigma Aldrich, St. Louis, MO, USA). The heart was cut from the cannula, atria removed, and placed in ~5–10 ml of formalin. After a minimum of 2 h, the heart was sliced into 5 equal sections using a Zivic Heart Slicer and returned to the formalin for 24 h before photographs were taken.

The stained heart sections were then photographed using the Leica S6D microscope and EC3 camera and associated LAS v4.12 software (Leica Microsystems, Buffalo Grove, IL, USA). Percent infarction was calculated by dividing the area of infarcted myocardium (white area) by the total area of myocardium (white+bright red area). The sum of these areas from each slice was used so that the reported percentage would best represent the

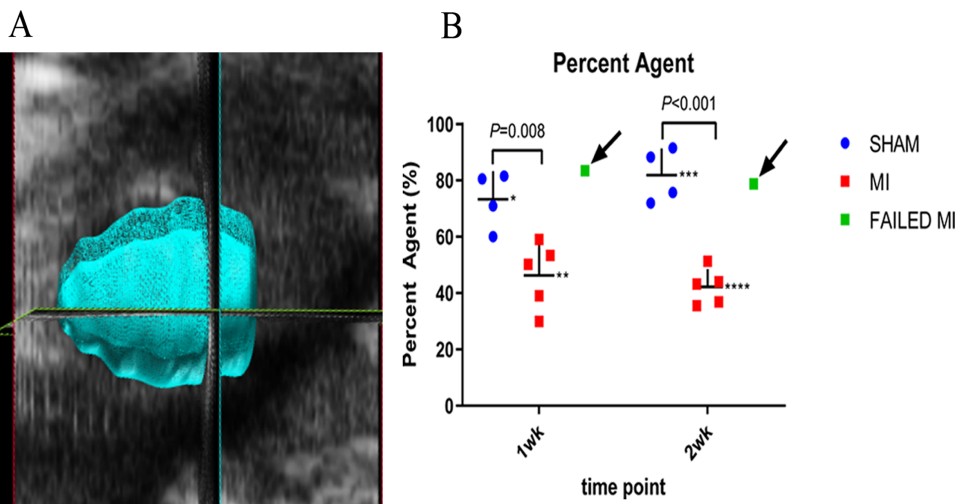

**Figure 2 Example image of a 3D myocardial reconstruction and associated PA data.** (A) 3D reconstruction of a sham-operated animal indicating the method used to define the myocardial region of interest (ROI) for evaluation of the MCE PA parameter. (B) Scatter plot of individual animal PA data. Data represents 1-week and 2-week cohorts of sham-operated animals (blue, $n = 4$), MI animals (red, $n = 5$), and failed MIs (green) with arrows indicating if they were able to be differentiated from the MI cohort. The failed MIs were excluded from mean data calculations. *73.27 ± 10.02, **46.32 ± 11.70, ***81.89 ± 9.49, ****42.17 ± 6.34. ROI, region of interest; PA, percent agent; MI, myocardial infarction.

amount of infarction of the entire left ventricle, enabling comparison of this value with the 3D PA.

## Statistics

All statistics were analyzed using GraphPad Prism® version 7.0 (GraphPad Software, San Diego, CA, USA). Power calculations to determine sample size were performed based on change in EF of MI versus sham-operated animals of surgeries performed previously in our hands. Using an alpha of 0.05 with 95% power, power calculation revealed that an n of 2 animals per group were sufficient to detect statistical significance. However, we utilized an n of 4–6 per group/cohort. Each conventional and MCE imaging technique was used to compare the MI animals to the sham-operated animals using multiple t-tests. Pearson correlations between each MCE parameter and the TTC histological standard were made with linear regression. Sham-operated and MI animals in their respective time cohort were combined in the correlation graphs to assess how the technique correlates with TTC staining in a range of degree of MI. A (two-tailed) *P* value of less than 0.05 was considered significant and all data are represented as mean ± standard deviation. For the data represented in Figs. 1–4, animals which failed to have infarctions as determined by TTC staining were removed from mean calculations. These scatter plots include the failed MIs as a green data point to assess if each technique could differentiate them from the rest of the MI cohort.

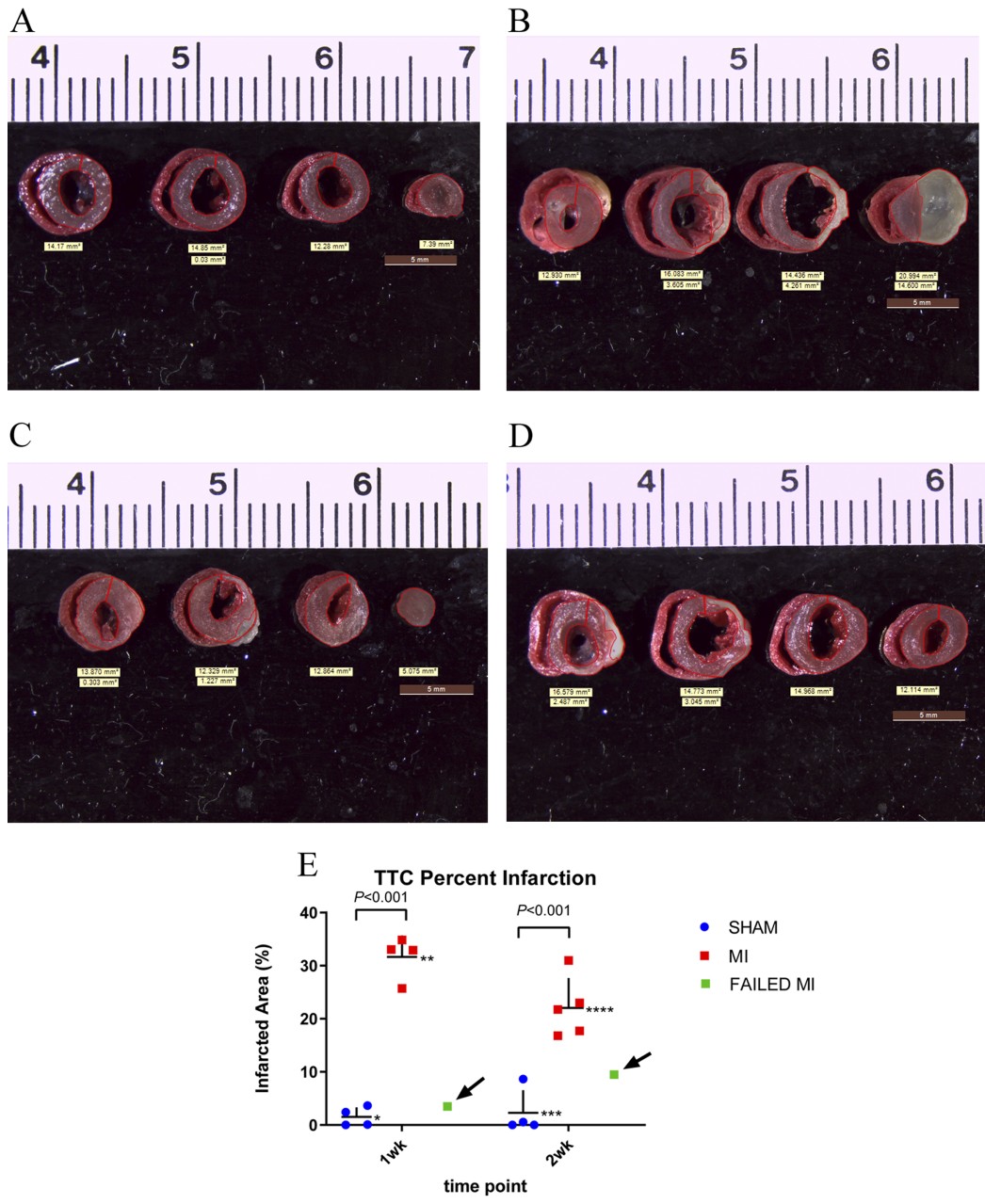

**Figure 3 TTC stained hearts collected at endpoint.** (A) Non-infarcted sham-operated animal showing normal perfusion throughout the myocardium as indicated by the bright red color. (B) MI heart showing non-perfused areas as indicated by the white color. (C & D) TTC staining of myocardium which failed to have a significant MI from 1-week and 2-week cohorts, respectively. (E) Individual animal data representing the TTC% infarction. The MI animals (red), sham-operated animals (blue), and failed MI animals (green) are depicted in 1-week and 2-week cohorts separately. The arrows indicate the failed MI animals that were able to be differentiated from the MI cohort. $n = 4$ for both sham-operated and MI animals in the 1-week cohort due to the destruction of one MI heart during histology preparation. $n = 4$ for the sham-operated and $n = 5$ for the MI animals in the 2-week cohort. The failed MIs were excluded from mean data calculations. *1.53 ± 1.80, **31.64 ± 4.04, ***2.30 ± 4.24, ****22.04 ± 5.63. TTC, triphenyltetrazolium chloride staining; MI, myocardial infarction.

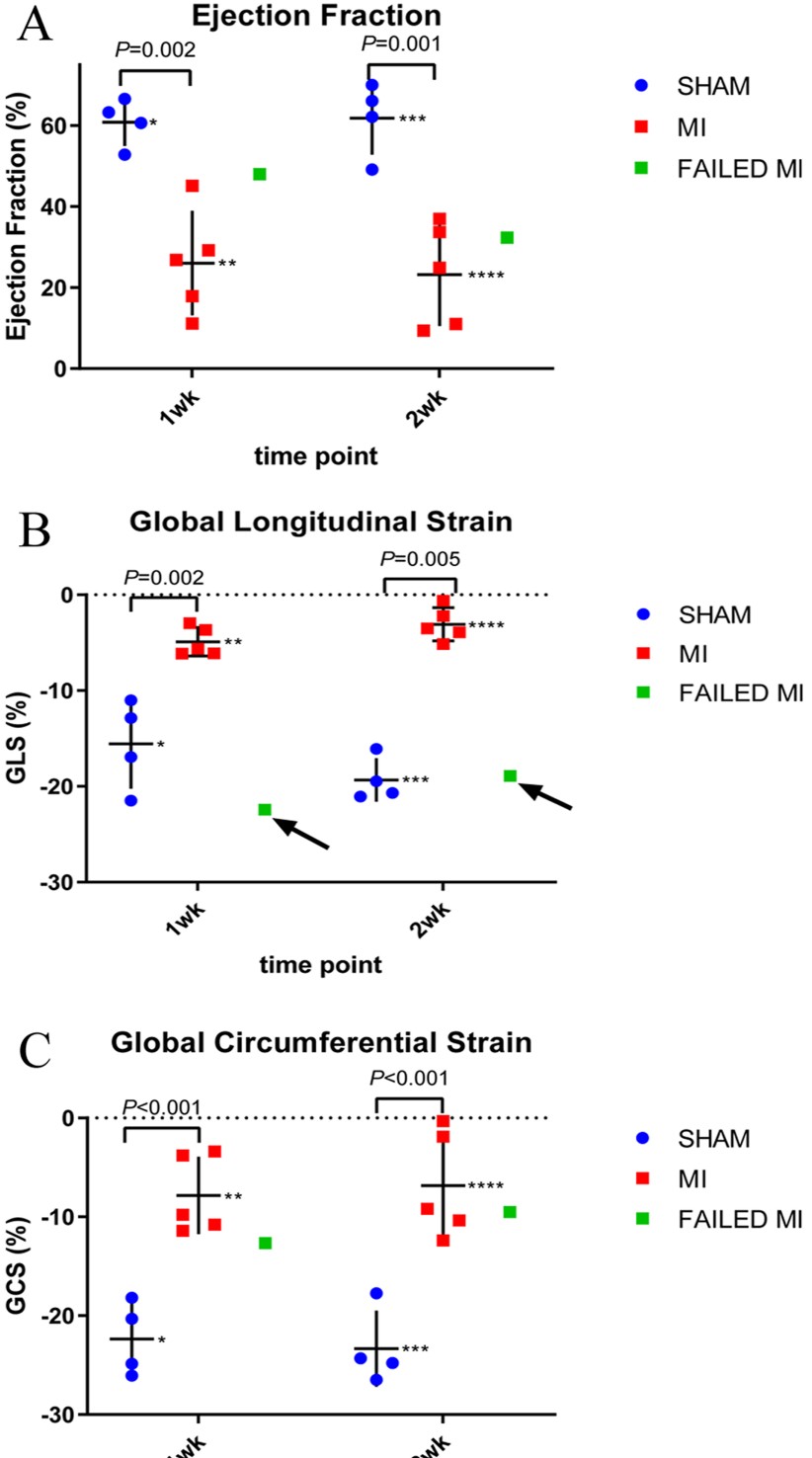

**Figure 4 Scatter plots of individual animal data from conventional echocardiogram techniques of MI (red, *n* = 5), sham-operated (blue, *n* = 4), and failed MI animals (green) in 1-week and 2-week cohorts.** The arrows indicate when a failed MI was able to be differentiated from the MI cohort. Failed MIs were not included in mean data calculations. (A) Represents the EF calculated from standard M-Mode. * 60.86 ± 5.86, ** 26.05 ± 12.89, *** 61.87 ± 9.06, **** 23.20 ± 12.66. (B) Represents the GLS as
**Figure 4** (continued)
calculated from 2D strain analysis. *−15.57 ± 4.86, **−4.90 ± 1.48, ***−19.33 ± 2.27, ****−3.08 ± 1.73. (C)
GCS as calculated from 2D strain analysis. *−22.35 ± 3.71, **−7.83 ± 3.92, ***−23.33 ± 3.85, ****−6.83 ±
5.38. MI, myocardial infarction; M-Mode, motion-mode; EF, ejection fraction; GLS, global longitudinal
strain; GCS, global circumferential strain.  

## RESULTS

### Animal model and injection technique

Standard LAD ligation surgery to induce MI was utilized to test the proposed MCE
imaging methods. The LAD ligation surgery resulted in a 71% survival of animals, giving a
total of 20 mice included in this study, resulting in $n = 4$–6 animals per group and
cohort. One animal in the 1-week cohort failed to have an MI and one animal in the 2-
week cohort had a very mild MI as determined by TTC staining (Figs. 3C and 3D).
These failed MIs were therefore removed from comparison of MI to sham-operated
animals in terms of mean data analysis in Figs. 1–4. However, note that the two failed MI
animals were included in correlation calculations and graphical analyses to demonstrate
the ability of each MCE parameter to correlate with TTC over a wide range of infarction
sizes. The retro-orbital injection was performed successfully in all animals and with
relative ease after mastery of the technique. There was damage to the globe of the eye in
only one animal, upon our first attempt of the injection technique.

### TTC staining results

TTC staining of the myocardium was performed in the 1-week and 2-week cohorts at
endpoint to enable comparison of this gold-standard infarction quantification method
with the proposed MCE parameter. TTC staining of the myocardium was successful in
all except one mouse which was excluded in all statistics. Figures 3A and 3B are
representative images of TTC stained sham-operated and MI hearts. Figure 3E depicts
the mean values of the percent infarct of the total myocardium (TTC% infarction).
The two failed MIs and the failed TTC stained heart were removed from these mean
calculations. For the 1-week cohort, MI ($n = 4$) and sham-operated animals ($n = 4$) resulted
in a TTC% infarction of 31.64 ± 4.04% and 1.53 ± 1.80% ($P < 0.001$), respectively.
For the 2-week cohort, MI ($n = 4$) and sham-operated animals ($n = 5$) resulted in a
TTC percent infarction of 22.04 ± 5.6% and 2.30 ± 4.24% ($P < 0.001$), respectively.
A scatter plot of these data, which include and indicate the failed MIs as a green color, is
seen in Fig. 3E.

### Conventional echocardiogram results

Echocardiograms to collect conventional functional data were performed in the 1-week
and 2-week cohorts to further compare to the proposed MCE technique parameters.
Echocardiographic data of M-mode EF is depicted in Fig. 4. Again, the two failed MIs were
excluded from mean calculations giving $n = 4$ of the sham-operated cohort and $n = 5$ of the
MI cohort. In the 1-week cohort, MI and sham-operated animal M-Mode EFs were 26.05 ±
12.89% and 60.86 ± 5.86% ($P = 0.002$), respectively. For the 2-week cohort, MI and

sham-operated animal M-Mode EFs were 23.20 ± 12.66% and 61.87 ± 9.06% ($P = 0.001$), respectively. A M-Mode EF scatterplot is represented in Fig. 4A and the failed MIs are included in this figure. Figures 4B and 4C represent strain analysis data of GLS and GCS, again with the failed MIs included in the scatterplot. Note how M-Mode EF and GCS could not specify the failed MIs; however, GLS measurements identified both failed MIs as indicated with arrows. The mean data from these measurements are seen in Fig. 4 with the failed MIs excluded from the means.

## MCE parameters

Image quality was good and remained reasonably consistent with all MCE images after optimal presets were determined (Fig. 1). PE, WiR, and PA mean values using nSIMB with standard deviations are seen in Figs. 1 and 2.

The 2D NLC PE and WiR represent data from the initial microbubble kinetic arrival period with nSIMB and are depicted in Fig. 1. These parameters reflect relative blood volume (PE) and relative blood perfusion (WiR) of the myocardium. For the 1-week cohort, the PE for MI and sham-operated animals were 75.08 ± 27.23 and 442.29 ± 163.00 ($P = 0.002$), respectively. For the 2-week cohort, the PE for MI and sham-operated animals were 65.68 ± 23.61 and 270.39 ± 75.01 ($P < 0.001$), respectively. For the 1-week cohort, the WiR for MI and sham-operated animals were 70.32 ± 95.91 and 401.86 ± 140.44 ($P = 0.004$), respectively. For the 2-week cohort, the WiR for MI and sham-operated animals were 39.27 ± 26.07 and 111.30 ± 65.12 ($P = 0.056$), respectively. Figures 1G and 1H depict the WiR and PE scatterplots with the failed MIs included and indicated as a green color. Note that for the WiR, the failed MI could be detected in the 1-week cohort, but not in the 2-week cohort. However, for PE, the failed MI was undetectable in the 1-week cohort but was detectable in the 2-week cohort.

The 3D PA technique is proposed to provide a more complete picture of vascularity since it includes the microbubbles circulating in the entire myocardium, rather than just a 2D image plane as is collected with PE and WiR. The final PA value was calculated by performing a background subtraction of the pre-scan (pre-injection) image from the ~1–2-min post-scan (post-injection) image while the nSIMB were stably circulating. Fig. 2A shows an example image of the ROI of the myocardial space defined to perform the 3D reconstruction. The PA was determined from this 3D ROI and quantifies the total amount of circulating microbubbles within the entire myocardium. As shown in Fig. 2, the PA for the 1-week cohort for MI and sham-operated mice were 46.32 ± 11.70% and 73.27 ± 10.02% ($P = 0.008$), respectively. The PA for the 2-week cohort for MI and sham-operated mice were 42.17 ± 6.34% and 81.89 ± 9.49% ($P < 0.001$), respectively. Again, the failed MIs were excluded from these calculations. However, as can be seen in Fig. 2B, the PA MCE technique was able to detect both the failed MIs in both cohorts as indicated by the arrows.

## MCE correlations with conventional techniques

To further validate the proposed MCE parameters with conventionally accepted MI evaluation methods, the resultant data was compared using liner regression. Pearson

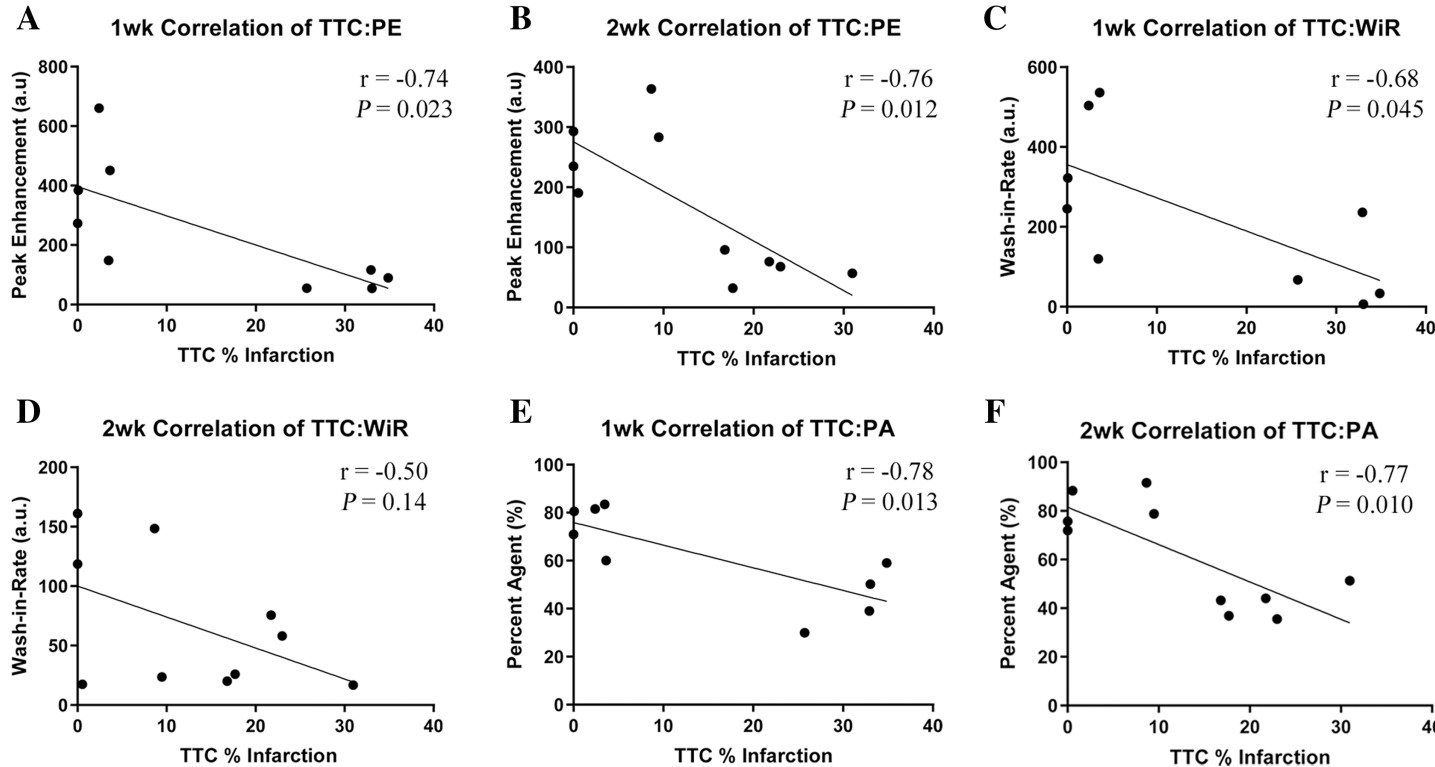

**Figure 5 Pearson Correlations depicting comparison of TTC staining versus each proposed MCE parameter at the 1-week and 2-week cohort with superimposed *r* and *P* values.** *n* = 4 for both sham-operated groups, *n* = 5 for the 1 week MI group and *n* = 6 for the 2 week MI group. (A) PE 1-week cohort. (B) PE 2-week cohort. (C) WiR 1-week cohort. (D) WiR 2-week cohort. (E) PA 1-week cohort. (F) PA 2-week cohort. MCE, myocardial contrast echocardiography; TTC, triphenyltetrazolium chloride staining; MI, myocardial infarction; PE, peak enhancement; WiR, wash-in-rate; PA, percent agent.

Correlations were plotted for TTC% infarction against each of the MCE parameters (PE, WiR, and PA) and for each cohort as shown in Fig. 5. Data from failed MI's were included in correlations. A 1-week MI heart was destroyed during its preparation for TTC staining, this heart's corresponding MCE parameters were excluded, resulting in *n* = 4 in both of the sham-operated groups, *n* = 5 in the 1-week MI group, and *n* = 6 in the 2-week MI group. The calculated r- and *P* values are shown in Fig. 5. Correlations of TTC% infarction versus PE in the 1-week cohort resulted in an r = −0.74 (CI [−0.94 to 0.14]) and in the 2-week cohort resulted in r = −0.76 (CI [−0.94 to 0.24]). Correlations of TTC% infarction versus WiR in the 1-week cohort resulted in r = −0.68 (CI [−0.93 to 0.025]) and in the 2-week cohort r = −0.50 (CI [−0.86 to 0.19]). Correlations of TTC% infarction versus PA in the 1-week cohort resulted in r = −0.78 (CI [−0.95 to 0.24]) and in the 2-week cohort r = −0.77 (CI [−0.94 to 0.26]). The only insignificant correlation was the WiR 2-week cohort (*P* = 0.14).

The three MCE parameters of PE, WiR, and PA were also correlated with the conventional functional parameters of 2D EF and GLS measured in PLAX. These results are seen in Table 2. All animals were included in these correlations giving an *n* = 4 in the sham-operated group and *n* = 6 in the MI group in both 1-week and 2-week cohorts. As before, all correlations represent strong relationships between the MCE parameters and

**Table 2 Pearson Correlations of conventional echocardiogram measurements of 2D EF and GLS versus the proposed MCE parameters of PE, WiR, and PA in the specified cohorts.**

| | 1 week 2D EF vs | | | 2 week 2D EF vs | | | 1 week GLS vs | | | 2 week GLS vs | | |
|---|---|---|---|---|---|---|---|---|---|---|---|---|
| | PA | PE | WiR | PA | PE | WiR | PA | PE | WiR | PA | PE | WiR |
| r | 0.77 | 0.76 | 0.67 | 0.86 | 0.85 | 0.57 | −0.80 | −0.65 | −0.53 | −0.88 | −0.88 | −0.52 |
| CI | 0.27 to 0.94 | 0.25 to 0.94 | 0.06 to 0.91 | 0.51 to 0.97 | 0.48 to 0.96 | −0.09 to 0.88 | −0.95 to −0.35 | −0.91 to −0.04 | −0.87 to 0.15 | −0.97 to −0.56 | −0.97 to −0.55 | −0.87 to 0.17 |
| P | 0.009 | 0.011 | 0.036 | 0.002 | 0.002 | 0.086 | 0.005 | 0.042 | 0.11 | <0.001 | <0.001 | 0.13 |

Notes:
$n = 4$ for all sham-operated cohorts and $n = 6$ for all MI cohorts.
MCE, myocardial contrast echocardiography; GLS, global longitudinal strain; 2D EF, two-dimensional ejection fraction percent; PE, peak enhancement; WiR, wash-in-rate; PA, percent agent.

the conventional functional parameters, with the exception of the WiR, which only correlated significantly with the 2D EF 1-week cohort ($P = 0.036$).

## Cationic vs neutral microbubbles

As stated, we performed the entire protocol in each animal with both nSIMB and cSIMB. The hypothesis was that the cSIMB would lodge in the myocardium vessels and enable improved imaging and quantification. Since the PE is the point at which the microbubbles have reached a stable plateau in circulation, it is proposed that if the bubbles were lodging in the myocardium successfully, then the PE value would be higher at the 8-min timepoint in the cSIMB than the nSIMB. To address this hypothesis, only sham-operated animals were included and 1- and 2-week cohorts were combined for n=8 for each cohort of nSIMB or cSIMB microbubble type. Table 1 depicts the PE value from the initial microbubble injection (PE Arrival) versus the PE at 8-min (PE 8-min) for both nSIMB and cSIMB. The PE at the plateau after initial injection (PE Arrival) was 356.34 ± 149.14 for the nSIMB and 354.58 ± 185.17 for the cSIMB with a $P$ value of 0.98, indicating no difference in lodging. After 8-min of plateau (PE 8-min), the PE was 49.35 ± 47.52 for the nSIMB and 24.64 ± 19.28 for the cSIMB calculating a $P$ value of 0.19. Although insignificant, this raises the possibility that there may have even been reduced lodging of the cSIMB. The reduction of cSIMB from 354.58 ± 185.17 at initial plateau to 24.64 ± 19.28 at 8-min plateau certainly provides no evidence of lodging of the cationic microbubbles. The WiR value for the initial microbubble injection arrival kinetics for each SIMB type is also included in Table 1, however, the differences in these values for the two types of microbubbles exhibited excess variability.

## DISCUSSION

This study provides proof of concept for a complementary technique to conventional TTC staining and echocardiography using noninvasive myocardial perfusion parameters. While these NLC tools have been primarily developed for the evaluation of the hemodynamic characteristics of tumors, we propose that they have applicability for evaluation of the heart. Overall, we successfully induced infarctions in all the mice assigned to the MI group, except two. Of these two fails, one did not exhibit any infarction and the other had an extremely mild MI. These two failed MIs were detected via TTC staining and conventional

functional imaging techniques. We then tested multiple MCE parameters collected using NLC that were able to successfully differentiate sham-operated vs MI animals. Importantly, these parameters correlated well with the gold standard TTC and conventional ultrasound functional imaging. Specifically, the 3D PA was able to detect both failed MIs and was well correlated with TTC at each timepoint. PE and WiR each detected one failed infarction. 2D B-Mode EF and GLS were also able to detect both failed MIs, however the commonly utilized, M-Mode EF and GCS were not.

The 3D PA value is thought to give information regarding the total vascularity of the myocardium as it indicates the percent of contrast agent present in the user-defined 3D region of interest. This is the parameter that most resembles the TTC technique as they are both 3D and derived from the perfusion of the myocardium. Just as TTC is less likely to stain the infarcted myocardium (*Klein et al., 1981*), the contrast agent is less likely to be present in the infarcted myocardium, resulting in less overall NLC signal transmitted from the microbubbles. In light of this, we advocate that the proposed MCE PA parameter can serve as a kind of "in vivo TTC." However, PA adds the benefit of providing in vivo perfusion data and can likely be used repeatedly throughout the experimental paradigm. Obvious value can be seen for this in models in which treatments of MI may be detected while in vivo at intermediate experimental timepoints, but otherwise missed at the endpoint. Additionally, this method may have utility in studies involving genetic manipulation of animals in which genes may be turned off and on, especially if the microvasculature is influenced. However, future studies will need to address whether NLC MCE imaging of the myocardium will be sensitive enough for subtler models of heart disease.

The PE parameter was statistically significant in differentiating sham-operated vs MI animals at both time points and correlated well with conventional methods. The WiR was statistically significant in its ability to differentiate sham-operated vs MI animals and correlated with TTC staining with the 1-week cohort, but not with the 2-week cohort. With our study's findings of the MCE parameters differentiating sham-operated vs MI animals and correlating significantly with gold standards, it is proposed that these values can provide valuable additional hemodynamic information regarding the perfusion of the myocardium. We speculate in our study that the PE value represents the relative blood volume within the myocardium. Our findings indicate this as lower PE values were observed in animals with more severe myocardial infarctions resulting in less perfusion and lower number of microbubbles present within the myocardium to emit a NLC signal. Myocardial blood volume has been shown to be useful to the researcher, correlating with functional aspects of coronary artery disease such as myocardial oxygen consumption (*McCommis et al., 2010*). The area under the curve value was also calculated by the software and could be considered; however, PE was chosen for this study as area under the curve best represents vascularity in tissue that has complex feeding vessels, such as tumors (*Greis, 2011*). The PE value is limited to providing information in a 2D plane; however, it provides the advantage of giving relative blood volume from the total NLC signal in that 2D image.

The WiR parameter resulted in the least amount of significant data of the MCE parameters, likely due to variation in microbubble injection rate. It needs to be investigated further to determine its value, given these results. Since WiR represents the rate of blood entering the myocardium, it is assumed to indicate the perfusion of the myocardium. This is one of the key advantages of MCE because it can provide information about micro-vessels and collateral vessel development, which is lost in other imaging techniques (*Kaul, 2008*). Myocardial perfusion with other modalities has an established role in research with calculating myocardial flow reserve, enhancing the researcher's ability to assess coronary artery disease (*McCommis et al., 2010*). The destruction-reperfusion technique may provide more accurate initial myocardial perfusion data as it eliminates the effects of the microbubbles having to traverse from the injection site to the myocardium (*Greis, 2011*). However, the bolus technique was chosen due to the lower volumes required for microbubble administration and increased ease for the researcher. Mice with MI commonly exhibit heart failure which would be further exacerbated by the volumes required from continuous infusions. Also, the ease of the bolus technique makes it more efficient for researchers to implement as it doesn't require an infusion pump and has been shown to have a higher contrast signal-to-noise ratio (*Averkiou et al., 2020*). Perhaps in mouse models where the systolic function is largely preserved, the destruction-reperfusion technique could be used to provide more accurate WiR data.

Our findings regarding M-Mode EF are consistent with previous studies that have compared these techniques and found limitations for the use of M-Mode in MI (*Chen et al., 2016*). M-Mode has the advantage of being fast and relatively easy to perform and may be used to assess ventricular function when studying non-coronary artery disease or animals that do not have regional wall motion abnormalities. Still, M-Mode assessment and the use of the Teichholz formula to calculate EF in mice are widely utilized (*Benavides-Vallve et al., 2012*). However, our observed limitations with M-Mode in assessing animals with MIs likely reflects why it is not recommended in clinical practice (*Lang et al., 2006*). The 2D B-Mode EF parameter is much more likely to account for the infarcted portion of the heart. The accuracy of our GLS measurements to detect failed MIs reflects previously validated STE imaging in mice (*Bauer et al., 2011*). GLS in clinical practice is continuing to develop and is currently recommended with patients receiving chemotherapy (*Plana et al., 2014*; *Smiseth et al., 2016*). The inability of the GCS measurement to detect the failed/very mild MIs likely reflects the same limitation seen with M-Mode in that it is assessing a single PSAX plane that may miss analyzing a more distal infarcted region. Although STE will likely gain utility and M-Mode may continue to have an application, these techniques are still limited.

The retro-orbital catheter injection technique is a valuable method for injecting microbubbles. Researchers often use the lateral tail vein for microbubble injections which can be difficult to perform in anesthetized animals exhibiting peripheral vasoconstriction or those with dark tail-colors. Retro-orbital injections are relatively easy to perform and shown to deliver the injected substance with the same kinetics as if it were injected in the tail vein (*Wang et al., 2015a*; *Yardeni et al., 2011*). Our modifications to the catheter, as described in the methods section, allowed us to keep the catheter inserted
in the retro-orbital sinus to facilitate multiple injections. We perforated one mouse eye during the initial attempt; however, all subsequent injections were performed successfully and efficiently by a novice injector.

We initially attempted to lodge neutral microbubbles within the myocardial vasculature as previously demonstrated (*Kaufmann et al., 2007*), with the goal of being able to visibly identify segmental perfusion defects without shadowing from microbubbles present in the ventricular lumen. Therefore, we trialed nSIMB 3–4, 4–5, and 5–8 μm from the manufacturer, hypothesizing that the larger bubbles may lodge. However, while the data is not presented here, we were not able to reproduce the lodging of nSIMBs in the myocardium with a ventricular lumen clear of microbubbles. Reasons for this may include differences in the composition of the microbubbles (i.e., different manufacturers) or differences in technique from previous publications. We also attempted to perform this lodging with cationic microbubbles, cSIMB 3–4 μm, due to previous reports that the cation charge may adhere to negatively charged endothelium (*Sirsi et al., 2012*). Our data indicated no significant evidence of lodging of cationic microbubbles. We were also unable to detect differences between nSIMB and cSIMB using the kinetic parameters seen in Table 1. However, the cSIMB measurements were performed after the nSIMB, possibly resulting in shifting of the animal or cardiac structure and more error with these measurements.

We determined that the nSIMB 3–4 μm gave us the best signal and after several trials to find the optimal microbubble concentration, we were able to improve the quality of images with reduced shadowing artifact from ventricular lumen microbubbles. Shadowing artifacts are still a limitation of the MCE technique, including those caused by the rib and lung, common to all mouse echocardiograms. Shadowing can also result from the contrast present in the tissue surrounding the heart along with potential scar tissue in the chest wall from MI surgery. Finally, the pericardium was noted to intermittently produce artifact that may have been detected as a false NLC signal, possibly reducing the accuracy of the WiR parameter.

A limitation often described with MCE is the lack of harmonic imaging associated with the high frequencies required for imaging mice. We noted a better signal with the size 3–4 μm microbubbles and this may be attributed to smaller bubbles producing more NLC signal with higher frequency transducers (*Goertz et al., 2003*). Although harmonic imaging may be reduced at these frequencies, we were able to detect enough NLC signal to perform these various MCE parameters, which has also been described elsewhere (*Raher et al., 2007*). This NLC imaging provides an advantage by filtering out other "noise" in the myocardium so that only microbubbles present in the vasculature is detected, giving a more accurate assessment of perfusion.

Other studies have described various techniques for quantifying myocardial perfusion/ MI in mice (*Alvarez et al., 2018*; *French et al., 2006*), but they do not quantify the NLC signal itself nor use readily available equipment and software as presented here. Rather, their measurements solely utilized pixel intensity and specialized software that may not be readily available or add complexity to a researcher's practice. Our study also improved on

these by offering multiple time points and comparing the techniques to TTC staining and conventional functional measurements.

While the power calculation indicated adequate sample size for statistical power, a major limitation of the present study includes the small sample size. Comparisons of the presented methods could be affected by variance in body weight, heart rate, and respiratory rate which may alter the metabolism of the microbubbles. The anesthetic regimen should be chosen and monitored carefully considering its effects of cardiovascular physiology and reported influence on echocardiographic parameters (*Wang et al., 2015b*). However, isoflurane anesthesia is the most widely used anesthetic for evaluation of mouse echocardiogram and thus was chosen in this study. Other limitations include that the user who performed the scans was new to mouse echocardiography and likely more prone to errors such as apical foreshortening when scanning. However, this may be considered an advantage that a novice user can readily perform this technique. The technique and measurements may have been performed with more accuracy as the user improved. When injecting the microbubbles, resistance can cause microbubbles to burst while in the syringe, altering the concentration of microbubbles actually injected. We addressed this by taking extreme care that the catheter once made was free moving by testing it with saline. Finally, in terms of logistics, gating for the heart rate in severely infarcted animals without obvious QRS complex occasionally failed to detect the ECG signal, resulting in movement of the heart and a displaced ROI.

Perhaps the largest limitation of this study was the inability to visibly distinguish infarcted from non-infarcted zones on 2D echocardiography. This may be due to inadequate resolution of the small mouse myocardium, especially after excessive thinning of the myocardial wall post infarction. Without this direct visualization of the perfusion defect, whether the tested parameters truly reflect infarction will need further study. It may be worth considering that new developments into our understanding of angiogenesis suggests that perfusion in MI zones may not be completely absent (*Kobayashi et al., 2017*). Perhaps with newer technology, including newer versions of ultrasound hardware and software, this resolution could be improved allowing segmental perfusion analysis and better characterization of the perfusion in the border of, and within, the infarction. Additionally, a report of the utility of the NLC system for the evaluation of the myocardium may spark interest in the development of software expressly for the purpose. Magnetic resonance imaging provides many advantages over echocardiography regarding cardiac anatomy resolution and soft tissue analysis; however, this modality is not widely accessible to researchers currently and continues to have limitations due to the high heart rate and respiratory rate of a mouse (*Bakermans et al., 2015*).

## CONCLUSION

Our study evaluates the ability of various MCE kinetic parameters to identify and characterize MI in an in vivo mouse model. Using readily available equipment and software in concert with the most commonly utilized anesthetic and cardiovascular assessment methods, MCE enabled collection of additional data to add value to the researcher toolbox. The majority of the PE, WiR, and 3D PA parameters correlated well

with TTC staining and conventional functional echocardiography measures in 1-week and 2-week post-MI cohorts. Our findings suggest that these parameters may provide complementary perfusion/hemodynamic information to the functional parameters already established with standard echocardiograms. However, further investigation of MCE in mouse models of MI is required before widespread adoption of this technique into basic scientific research. This study provides the groundwork for further assessment of these parameters to determine their accuracy and role in the evaluation of murine MI.

## ACKNOWLEDGEMENTS

We would like to thank Dr. Doug Martin and Dr. Yi-Fan Li (University of South Dakota) for their assistance with this project and the preparation of this manuscript. We would also like to thank Jerad Schlobohm (South Dakota State University) for his help preparing the figures. Lastly, we would like to express appreciation for access to the equipment and services provided by the Physiology Core Facility in the Department of Basic Biomedical Sciences at The University of South Dakota.

### Funding

This work was supported by the Medical Student Research Program at the University of South Dakota. The funders had no role in study design, data collection and analysis, decision to publish, or preparation of the manuscript.

### Grant Disclosures

The following grant information was disclosed by the authors:
University of South Dakota.

### Competing Interests

The authors declare that they have no competing interests.

### Author Contributions

- Nicholas T. Thielen conceived and designed the experiments, performed the experiments, analyzed the data, prepared figures and/or tables, authored or reviewed drafts of the paper, and approved the final draft.
- Adison A. Kleinsasser analyzed the data, prepared figures and/or tables, and approved the final draft.
- Jessica L. Freeling conceived and designed the experiments, performed the experiments, analyzed the data, authored or reviewed drafts of the paper, and approved the final draft.

### Animal Ethics

The following information was supplied relating to ethical approvals (i.e., approving body and any reference numbers):

The University of South Dakota Institutional Animal Care and Use Committee approval was obtained for all experiments (IACUC # 12-12-17-20D).

## Data Availability

The raw measurements are available in the Supplemental File.

2D parasternal long axis video of microbubbles entering a MI induced heart from the 1-week cohort: Thielen, Nick; Freeling, Jessica; Kleinsasser, Adison (2021): MI mouse microbubble arrival clip. figshare. Media. DOI 10.6084/m9.figshare.12977696.

2D parasternal long axis video showing microbubbles enter a sham-operated heart from the 1-week cohort: Thielen, Nick; Freeling, Jessica; Kleinsasser, Adison (2021): Sham-operated mouse microbubble arrival clip. figshare. Media. DOI 10.6084/m9.figshare.12977672.v1.

## Supplemental Information

Supplemental information for this article can be found online at http://dx.doi.org/10.7717/peerj.11500#supplemental-information.

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
