# Peer review of "Myocardial contrast echocardiography assessment of mouse myocardial infarction: comparison of kinetic parameters with conventional methods"

_PeerJ, doi:10.7717/peerj.11500_

## Round 0.1 · original submission · Minor Revisions

As you gather from reviews, the reviewers identified several methodological, legibility and language issues and I second that. It is important to revise the manuscript per reviewers' comments to make it an engaging read and further editorial consideration.

·

Basic reporting

The introduction was very long, and sometimes it is not easy to follow. A good introduction should identify the article topic and provide essential context. I suggest rewriting the introduction in brief.

The conclusion should be more specific and has conclusive statements. Please rewrite!

The figures should be including the (P values) between the MI and sham groups to become more understandable for the readers.

Experimental design

'no comment'

Validity of the findings

It was the first time that compared the myocardial contrasts echocardiography (MCE) with the speckle tracking data in the small animal model. However, the evaluation of MCE by the gold standard (TTC stain) was previously reported in Bhan et al. 2013 (doi:10.1152/ajpheart.00553.2013)

Results. In fact, It could not understand why the data of myocardial infarction( MI) mice and sham were mixed and presented as one group (correlations figures). The Author aimed to demonstrate the usefulness of the MCE method with MI. Thus, it could not plot the data as one group, even though the study sample was very small. Please comment.

Additional comments

Many limitations should be considered and mentioned.
The first and major limitation should be the groups' sample size, which can affect comparisons of small differences between methods. Therefore, the results should be confirmed in a larger series.
When assessing the left ventricular function, an anaesthetic regimen should be chosen carefully to minimize its effect on cardiovascular physiology. The adverse effects of anaesthesia on conventional and speckle tracking echocardiographic parameters have been previously reported.
Moreover, the apex foreshortening should be considered. The apex foreshortening could result from the chest anatomy of animal models, surgical operation, and more frequently in acquisitions obtained from less experienced sonographers. The effect of apex foreshortening on conventional and strain measurements should be stated as a limitation.

·

Basic reporting

- The English was overall clear and unambiguous.
- The literature was sufficiently referenced
- The raw data set was shared.
- Self-contained with relevant results to the hypothesis
Relevant comments are given in the general comments to the authors.

Experimental design

Original, rigorous, well-defined research using methods with sufficient detail.

Validity of the findings

Robust, statistically sound data.

Additional comments

The authors describe a readily available echocardiography tool to non-invasively measure blood volume and perfusion after myocardial infarction, and validated it with conventional methods.

Overall the authors have a well -written manuscript with solid scientific data using well-described methods.

There are a few issues that should be addressed:

Major issues:
The authors should clarify better the usefulness of measuring noninvasively blood volume and perfusion, and what it could be used for in research.

Minor comments:
Abstract:
-The first sentence in the abstract should be simplified/ split in two:
“This study explores the use of and provides proof of concept for a minimally invasive assessment of myocardial infarction in mice using myocardial contrast echocardiography (MCE) and existing equipment and software readily available to the average researcher. “
- the abbreviations that are not used more than two times can be taken out (this is also true for the manuscript)
Manuscript:
-p1 line 35-36: Also small animal MRI is being used. – please add
- Table 1 is redundant if all the means and p values were added to the graphs.

- Figures: now from only looking at the graph, I understand the method does not work, therefore: in addition to the arrows to the data point of the failed MI, please separate the failed MI point with a different color (but same shape) and an additional legend in the figure. For the statistical test state that this point was excluded. If possible, add the TTC for this datapoint.

Figure 1: Point out in graph 1C &1F, point out where wash in rate and peak enhancement were taken from.

Line 357 An M-mode: should be: “A M-mode..”

Line 158: polyethylene PE 20 tubing : PE is confusing in this sentence, please erase – as later you use it for peak enhancement.
Line 156: “proof of concept for an innovative complementary technique to conventional TTC staining and echocardiography”: add noninvasive / perfusion measurements to the sentence
-Discussion: add noninvasive MRI (dynamic contrast enhanced/ late gadolinium enhanced) to the discussion. E.g. https://pubmed.ncbi.nlm.nih.gov/26282195/

---

## Round 0.2 · accepted · Accept

Thanks for considering the reviewers' comments and improving the manuscript.

·

Basic reporting

No comments

Experimental design

No comments

Validity of the findings

No comments

Additional comments

No comments

·

Basic reporting

no comment

Experimental design

no comment

Validity of the findings

no comment

Additional comments

The authors have addressed all issues raised.